# *lil*Gym: Natural Language Visual Reasoning with Reinforcement Learning

**Anne Wu, Kianté Brantley, Noriyuki Kojima, and Yoav Artzi**

Department of Computer Science and Cornell Tech, Cornell University

`{aw588, kdb82, nk654}@cornell.edu, {yoav}@cs.cornell.edu`

## 1   Introduction

Reinforcement learning (RL) with natural language context poses important opportunities and challenges. Language provides an expressive and accessible conduit for task specification, so that RL agents can address a broad set of tasks, rather than learn a single behavior. For natural language processing (NLP), RL is a promising avenue for language use and acquisition with world interaction. Fulfilling this potential requires addressing core reasoning challenges: the RL agent must reason about both high-level language concepts, low-level actions, and the relations between them.

Despite significant interest and promising approaches, it has been challenging to create expressive RL benchmarks with natural language. Existing approaches often make various simplifications, such as using synthetic language [Côté et al., 2018, Co-Reyes et al., 2019] or heuristic approximation, for example via demonstration data [Misra et al., 2017]. While these approaches open new avenues for research, they either do not explore the full complexity of natural language or introduce unexpected artifacts into learning through meaning approximations.

We present *lil*Gym,[1] a reinforcement learning benchmark for natural language visual reasoning that addresses the above issues. It includes the semantically diverse natural language of the Natural Language for Visual Reasoning (NLVR) corpus [Suhr et al., 2017], which has highly compositional human-written language and requires complex grounded reasoning. *lil*Gym provides an executable evaluation function for every statement in the NLVR corpus, and these annotations align the reward function with the language semantics of the agent's underlying reasoning task. We experiment with standard on-policy RL algorithms. Our experimental results show that while existing methods are able to achieve non-trivial performance, the complex visual reasoning required by *lil*Gym forms a challenging open problem.

## 2   The *lil*Gym Benchmark

*lil*Gym consists of a collection of environments that share a common backbone. The backbone is a 2D plane that is manipulated by placing and removing objects of different types. Each environment instance is a Markov Decision Process (MDP) created by pairing a natural language statement and a target boolean value with a configuration of the shared backbone. The goal of the agent in each environment is to manipulate it by adding and removing objects so that the truth-value of the statement with regard to the environment is the target boolean.

The learning problem *lil*Gym presents is to induce a policy that generalizes across MDPs. We split the MDPs to training, development, and held-out testing sets. The training environments are to be used for parameter estimation, while the two other sets are for testing during development and for final held-out testing to report approach performance.[2]

---

[1] *lil*Gym stands for Language, Interaction, and Learning Gym.

[2] We recommend reporting both development and held-out test results in future work for easy comparison.

36th Conference on Neural Information Processing Systems (NeurIPS 2022).

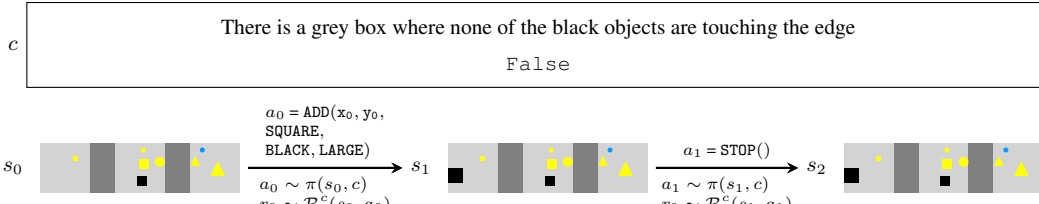

Figure 1: Overview of an example for one CMDP, SCATTER-FLIPIT. The context $c$ consists of a text statement and a target boolean. The leftmost image depicts the initial state $s_0$. The agent $\pi$ is presented with $(s_0, c)$, and chooses an action $a_0 \sim \pi(\cdot|s_0, c)$. The environment transitions to the next updated state $s_1$, the context remaining the same.

There are two dimensions of configuration: appearance and starting condition. The appearance determines the state space, transition function, and action space. The appearance of the environment can be (a) TOWER: the objects include squares only, and they can be stacked into towers in specific positions only; or SCATTER: objects of different types can be freely distributed. The starting condition determines the agent's goal. The starting condition and agent objective can be: (a) SCRATCH: the environment starts without any objects and the goal is to modify it so that the statement's truth-value is True; or (b) FLIPIT: the environment starts with a set of objects and the agent's goal is to flip the truth-value of the statement.

There are four configurations: TOWER-SCRATCH, TOWER-FLIPIT, SCATTER-SCRATCH, and SCATTER-FLIPIT. Each configuration forms a Contextual Markov Decision Process [CMDP; Hallak et al., 2015]. CMDP is an abstraction over a set of MDPs to account for a context that remains constant throughout the interaction with an MDP. We set the context to include the statement and the target boolean the interaction is conditioned on. A CMDP is a tuple $(\mathcal{C}, \mathcal{S}, \mathcal{A}, \mathcal{M}(c))$, where $\mathcal{C}$ is the context space, $\mathcal{S}$ the state space, $\mathcal{A}$ the action space, and $\mathcal{M}$ a function mapping a context $c \in \mathcal{C}$ to an MDP $\mathcal{M}(c) = (\mathcal{S}, \mathcal{A}, T, R^c, \beta^c)$. Here, $T : \mathcal{S} \times \mathcal{A} \to \mathcal{S}$ is a transition function, $R^c : \mathcal{S} \times \mathcal{A} \to \mathbb{R}$ the reward function, and $\beta^c$ an initial state distribution. This means that a CMDP is a set of MDPs that share the same states and actions. Table 2 in Appendix A shows the number of MDPs under each configuration. The policy takes as input both the current state and the context that created the MDP. The learning problem is to estimate parameters $\theta$ for a policy $\pi_\theta : \mathcal{S} \times \mathcal{C} \to \mathcal{A}$.

**Contexts** A context $c \in \mathcal{C}$ is a pair $c = (\bar{x}, b)$, where $\bar{x}$ is a natural language statement and $b \in \{\text{True}, \text{False}\}$ is a target boolean value for the statement $\bar{x}$ with respect to the state $s$. The target boolean value in SCRATCH is always True. In FLIPIT, the target boolean value can either be True or False.

**States** A state $s \in \mathcal{S}$ is an RGB image. Images in *lil*Gym are divided into three box regions of identical dimensions by two gray separators (Figure 1). The objects in *lil*Gym have three properties, each can take multiple values: shape (CIRCLE, SQUARE or TRIANGLE), color (BLACK, BLUE, or YELLOW), and size (SMALL, MEDIUM or LARGE). In TOWER, states are constrained to have stacks of up to four SQUAREs of MEDIUM size and any color at the center of each box. SCATTER states support all object shapes, sizes, and colors, and they may be positioned freely. In both conditions, objects cannot cross image boundaries or into the separators. The choice between SCRATCH or FLIPIT does not influence the state space.

**Actions and Transitions** There are three action types STOP, ADD, and REMOVE. STOP terminates the episode and does not require any parameters. The truth-value of the statement is only evaluated and compared to the target boolean after the STOP action is taken. ADD adds objects to the environment, and REMOVE removes objects. They take different arguments for TOWER and SCATTER:

**TOWER:** Similar to the state space of TOWER, the actions are also more constrained. Both ADD and REMOVE take a position argument, which has three possible values corresponding to the three box regions. Objects are always added or removed at the top of the stack. Adding an object on top of a stack of four objects or removing an object from an empty box are both invalid actions. ADD also takes a color argument. Including STOP, there are $1 + (3 + 1) \times 3 = 13$ actions.

**SCATTER:** Unlike TOWER, objects of any type can be placed freely in the box regions. Both ADD and REMOVE take 2D coordinates that specify the pixel location. Adding an object places it so that its top-left coordinates are the given coordinates. Removing an object will remove the object at the given coordinates. Adding also requires specifying the shape, color, and size. If adding results in objects overlap or boundary crossing with the separators or image boundaries, the action is invalid. Removing from a position that does not include an object is also an invalid action. The native resolution of images is $380 \times 100$ pixels. Including STOP, there are $1 + (380 \times 100) \times ((3 \times 3 \times 3) + 1) = 1,064,001$ actions. In our experiments (Section 4), we use a grid of $19 \times 5$, giving a total number of 2,661 actions.

The environment transitions are controlled by the transition function $T : \mathcal{S} \times \mathcal{A} \to \mathcal{S}$. $T$ depends on the choice between TOWER and SCATTER, because this choice determines the action space. The transition function does not modify the context, which is fixed for a given MDP.

**Reward Function** The reward function $R^c$ is computed with respect to the context pair $c = (\bar{x}, b)$, and is based on evaluating the truth-value of the natural language statement $\bar{x}$ with respect to a state $s$, and comparing it to the target boolean $b$. *lil*Gym includes an executable evaluation function $\mathcal{E}^{\bar{x}} : \mathcal{S} \times \mathcal{A} \to \{\texttt{True}, \texttt{False}\}$ for every statement $\bar{x}$. We describe how we create these evaluation functions in Section A.3.

The agent receives a positive reward for terminating the episode using the STOP action with the statement evaluation $\mathcal{E}^{\bar{x}}(s)$ equal to the target boolean value $b$. If the statement boolean value $\mathcal{E}^{\bar{x}}(s)$ does not equal the target boolean $b$ value when taking the STOP action, the agent receives a negative reward. If the episode terminates because the current time step $t$ reached the action horizon $H$ or because of an invalid action, the agent also receives a negative reward. Action validity depends on the current state $s$ and on the configuration, because TOWER and SCATTER have different action spaces. There is also a verbosity penalty of $\delta$ for every other action. Formally, the reward is:

$$R^c(s, a) = \begin{cases} 1.0 & a = \texttt{STOP} \land \mathcal{E}^{\bar{x}}(s) = b \\ -1.0 & a = \texttt{STOP} \land \mathcal{E}^{\bar{x}}(s) \neq b \\ -1.0 & (a \text{ is invalid in } s) \lor (t = H) \\ -\delta & \text{otherwise} \end{cases} .$$

**Initial State Distribution** The initial state distribution $\beta^c$ is parameterized by the context $c \in \mathcal{C}$, which is different between SCRATCH and FLIPIT. In SCRATCH, the agent modifies an empty environment to satisfy the truth-condition of the statement $\bar{x}$ in the context $c$, so the initial state $s_0$ is always an empty image. The set of initial states $\beta^c$ for every context $c \in \mathcal{C}$ is the set of images associated with the statement $\bar{x}$ in the NLVR data. In practice, for FLIPIT, this set includes between 1 to 43 images.

## 3 Experimental Setup

We consider two RL algorithms paired with two models for a total of four algorithm-model pairs.

### 3.1 Models

In our experiments, we consider two models: C+BERT and ViLT. These two models learn a joint visuo-linguistic representation, which is necessary to solve the proposed CMDP configurations.

**C+BERT** We process the statement $\bar{x}$ using BERT [Devlin et al., 2019], and do mean pooling across all layers and tokens to get the statement representation. We use a three-layer CNN [Fukushima and Miyake, 1982] to embed the image of the current state $s$. We concatenate the statement representation, image representation, and an embedding for the target boolean $b$, and process the vector through a multi-layer perceptron (MLP) to compute the action distribution.

**ViLT** ViLT is a pretrained multi-modal Transformer that jointly processes text and image inputs [Kim et al., 2021]. We create a sequence of tokens by concatenating the statement, a token for the target boolean, and image patches, separated by special tokens. The image patches are the same size as the $19 \times 5$ grid cells, including in TOWER, where the action space does not use a grid.

Table 1: Accuracies for all the four CMDP, with both models (C+BERT and ViLT) and both algorithms (PPO and PPO+SF). Evaluation is done without stop forcing (i.e. with PPO).

| | | TOWER–SCRATCH | | TOWER–FLIPIT | | SCATTER–SCRATCH | | SCATTER–FLIPIT | |
|---|---|---|---|---|---|---|---|---|---|
| | | Dev | Test | Dev | Test | Dev | Test | Dev | Test |
| PPO | C+BERT | 71.78 | 63.27 | 35.95 | 34.78 | **39.08** | **48.39** | 0.00 | 0.00 |
| | ViLT | **81.60** | **76.54** | **67.60** | **65.80** | 35.63 | 41.29 | **3.51** | **6.09** |
| PPO+SF | C+BERT | 80.98 | 78.70 | 27.22 | 26.75 | **70.12** | **74.84** | 8.31 | 8.46 |
| | ViLT | **84.05** | **82.41** | **65.09** | **62.91** | 64.37 | 70.97 | **27.48** | **29.95** |

## 3.2 Algorithms

We use PPO [Schulman et al., 2017] for parameter estimation,[3] with a separate network as a critic. The critic network is identical to the policy, except that we add a `tanh` activation for the value output. Because of the large action space, especially for SCATTER, the agent rarely observes positive reward, which requires taking a STOP action at an appropriate state. We design a simple variant of PPO called PPO+SF (PPO with stop forcing) to study this issue. PPO+SF is identical to PPO, except that during training, we mask all actions except STOP when the agent reaches a state where selecting STOP will give a positive reward. This modification is present only during training. All testing is done under the same conditions, without stop forcing.

# 4 Experiments

For the experiments we focus on two questions.

**How do the algorithm-model pairs perform?** For every CMDP, we report accuracy on the development and test sets for both PPO and PPO+SF and both C+BERT and ViLT. For every configuration, we randomly sample 10% of the training data as a held-out validation set kept unchanged throughout the experiments. We stop training using this validation set with early stopping and select the model with the best validation accuracy.

In Table 1, we observe that the additional guidance of PPO+SF compared to PPO helps with exploration, especially on SCATTER CMDPs. On SCATTER–FLIPIT, PPO+SF improves performance by 23.86% compared to PPO. This illustrates the hard exploration problem that SCATTER CMDPs pose. ViLT generally outperforms C+BERT, except on SCATTER–SCRATCH. This is relatively expected given the joint reasoning architecture and multi-modal pre-training of ViLT.

**What types of mistakes do PPO models make?** We sample 50 erroneous development examples from configurations in the SCATTER–FLIPIT and SCATTER–SCRATCH CMDPs, and analyze their mistakes. In SCATTER–SCRATCH trained with PPO, we found that for C+BERT, 76% of the mistakes are due to invalid actions, and 24% due to early termination. Among the invalid actions, 58% are due to trying to put an item that cannot fit in the box, 24% are due to trying to perform an action on a separator, and 18% due to trying to remove an object from a position that does not include an object. For other configurations, the error causes are similar.

# 5 Conclusion

We introduce *lil*Gym, a reinforcement learning benchmark focused on natural language visual reasoning. *lil*Gym is designed to be accessible for RL researchers, while still displaying the reasoning richness of natural language. Our data annotation approach allows including expressive and diverse natural language, while still providing accurate and automatic reward computation. Our strong baselines show that existing methods can achieve non-trivial performance on *lil*Gym, but there remain significant challenges to be solved and progress to be made.

---

[3]We use the PPO implementation of Kostrikov [2018].

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

Table 2: Data statistics per CMDP configuration and data split. The number of MDPs corresponds to the number of contexts under each CMDP. For `FLIPIT`, "Init." corresponds to the total number of initial states across all MDPs for this CMDP.[4]

| | TOWER-SCRATCH | TOWER-FLIPIT | | SCATTER-SCRATCH | SCATTER-FLIPIT | |
|---|---|---|---|---|---|---|
| | MDPs | MDPs | Init. | MDPs | MDPs | Init. |
| Train | 989 | 1,910 | 5,704 | 1,241 | 2,340 | 6,696 |
| Dev | 163 | 317 | 676 | 87 | 164 | 313 |
| Test | 324 | 619 | 1,383 | 155 | 285 | 591 |
| Total | 1,476 | 2,846 | 7,763 | 1,483 | 2,789 | 7,600 |

# A  The *lil*Gym Data

The data used for *lil*Gym is based on the Natural Language for Visual Reasoning (NLVR) corpus [Suhr et al., 2017]. The NLVR data was initially collected as a supervised learning benchmark. We formalize an interactive task on top of the NLVR data and collect additional annotations for reward computation.

## A.1  Background: the NLVR Corpus

NLVR includes human-written natural language statements paired with synthetic images. Each pair is annotated with the boolean truth-value of the statement with regard to the image (i.e., `True` if the statement is true with regard to the image, or `False` otherwise). The images are designed to support complex reasoning, including about spatial and set relations. The original learning task posed by NLVR is to classify statement-image pairs as `True` to indicate the statement is true with regard to the image, or `False` otherwise. Various approaches were developed to address the NLVR challenge [Suhr et al., 2017, Tan and Bansal, 2018, Goldman et al., 2018, Pavez et al., 2018, Yao et al., 2018, Hudson and Manning, 2018, Perez et al., 2018, Dasigi et al., Zheng et al., 2020, Gupta et al., 2021], and a separate version using photos was also released [Suhr et al., 2019].[5] NLVR is roughly balanced, and the current state-of-the-art using the raw images is 80.6% accuracy [Zheng et al., 2020], leaving significant room for improvement and illustrating the reasoning challenges the NLVR data presents.

Qualitative analysis of the data [Table 2 in Suhr et al., 2017] shows a more diverse representation of semantic and compositional phenomena compared to related corpora [Antol et al., 2015], including requiring joint visual-linguistic reasoning about spatial relations, quantities, and sets of objects. NLVR also provides an underlying structured representation for every image, which supports easy image manipulation. The combination of a simple interface for image manipulation with complex reasoning via natural language makes NLVR ideal to support an interactive benchmark environment.

## A.2  Constructing *lil*Gym from NLVR

We use the NLVR data to create each of the CMDPs (Table 2). `SCRATCH` CMDPs include contexts for all natural language statements from NLVR, each paired with the empty initial state containing no shapes. `FLIPIT` CMDPs include the natural language statements with their corresponding images, both from NLVR (an example from `SCATTER-FLIPIT` in Figure 1). The images are used as initial states. The target boolean is set so that the initial state does not fulfil it.

The split between `TOWER` and `SCATTER` also follows from NLVR. Statements corresponding to `TOWER` images in NLVR are included in our `TOWER` CMDPs, and the same for `SCATTER` sentences.

---

[4]NLVR includes a total of 18,322 images. This allows further expanding the number of initial states to 92,179 initial states through box element permutations. We do not manipulate this property in this work, but future work could take advantage of it. Our reward computation is invariant to such permutations.

[5]We do not use the photographic NLVR2 in this work.

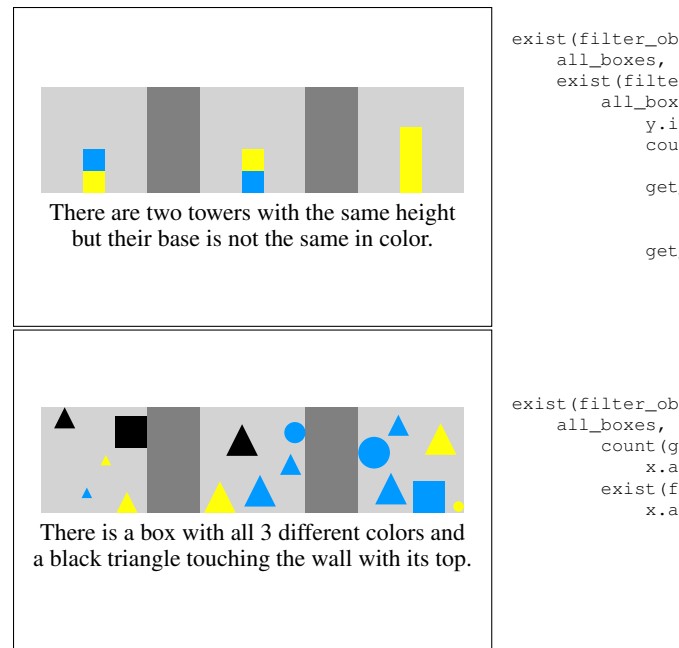

```
exist(filter_obj(
    all_boxes, lambda x: x.is_tower() and
    exist(filter_obj(
        all_boxes, lambda y:
            y.is_tower() and
            count(x.all_items_in_box()) ==
                count(y.all_items_in_box()) and
            get_set_colors(filter_obj(
                y.all_items_in_box(),
                is_bottom)) !=
            get_set_colors(filter_obj(
                x.all_items_in_box(),
                is_bottom))))))
```

There are two towers with the same height but their base is not the same in color.

```
exist(filter_obj(
    all_boxes, lambda x:
        count(get_set_colors(
            x.all_items_in_box())) == 3 and
        exist(filter_obj(
            x.all_items_in_box(), lambda y:
                is_black(y) and
                is_triangle(y) and
                is_touching_wall(y, Side.TOP)))))
```

There is a box with all 3 different colors and a black triangle touching the wall with its top.

Figure 2: Example sentences with the example images displayed alongside them during annotation (left), and their annotated Python program representation (right). Both sentences and logical forms are `True` for the corresponding image.

NLVR has four splits for training, development, public testing, and hidden testing. We follow the original splits for the training and development sets. Following the recent public release of the hidden testing set, we merge the public and hidden testing sets into a single public test split.

### A.3 Annotations for Reward Computation

The NLVR annotations include the truth-value of each statement with regard to the images paired with it in the data. Once we manipulate an image (i.e., change the state in our interactive environment), the truth-value annotation does not hold. A key challenge for creating an interactive environment using the NLVR data is the need for an accurate evaluation of the natural language statement for *every* possible state (i.e., image), as required for reward computation (Section 2).

We address this challenge by annotating each statement $\bar{x}$ with an executable boolean Python program representing its meaning, $\mathcal{E}^{\bar{x}}$ in Section 2. The Python programs operate on the underlying structured representation. Each program returns `True` for every image that satisfies the constraints specified in the corresponding statement, and `False` otherwise. In general, there are many states that satisfy any given statement, many more than provided with the original NLVR images.

The programs are written using an API defined over the structured representations. We base the API design on the logical ontology designed for NLVR's structured representations by Goldman et al. [2018], which we extend to include a total of 66 functions. Figure 2 shows two examples of logical forms paired with their corresponding statements.

We use the freelancing platform Upwork[6] for annotation. We recruit three annotators based on preliminary screening of their fluency in English and competency in Python. We de-duplicate the naturally occurring sentences in the data, collect 2,666 annotations at a total cost of $3,756, and keep 2,661 valid annotations.

All the sentences in the dataset are randomly distributed to the annotators, each with an example image. Every sentence is annotated with a logical form by one annotator. Each logical form is evaluated against a corresponding hidden validation set, and must pass all the tests.

---

[6] https://www.upwork.com

## B  Related Work

There is significant and increasing interest in RL conditioned on natural language. Various strategies are deployed to resolve language semantics for reward computation, mostly by strict control of the language or through approximations.

Maybe the most common approach is to control the language by using synthetic language backed by a formal representation [Narasimhan et al., 2015, Johnson et al., 2017a,b, Côté et al., 2018, Chevalier-Boisvert et al., 2019, Co-Reyes et al., 2019, Jiang et al., 2020]. Although synthetic language allows studying the problem of learning high-level concepts, many of the complexities of natural language are stripped away, and such approaches run the risk of reducing the language learning challenge to reverse engineering the hand-crafted generation process.

An alternative that allows for natural language while retaining the control of its semantics is to generate the target sequence of decisions (i.e., task demonstration), and solicit post-hoc instructional language [Shridhar et al., 2020, 2021, Hanjie et al., 2021]. While this process uses human-written language, it potentially implicitly retains the regularities of the demonstration generation procedure.

Others have carefully designed the underlying environment to simplify termination state evaluation given demonstrations, for example, with a sparse graph-based structure [Anderson et al., 2018, Chen et al., 2019, Ku et al., 2020]. However, recent work shows the potential for evaluation fidelity issues even in these settings [Jain et al., 2019].

In contrast to previous approaches, we emphasize using human-written natural language and avoid constraining the task to simplify reward evaluation. We also opt to not use underlying hand-crafted procedures as stimuli for the writing. *lil*Gym prioritizes exact reward computation rather than automated approximations to allow for relatively clean benchmarking of learning methods.

Our annotation of natural language statements with programs is inspired by the annotation of data for supervised learning of semantic parsers [Zelle and Mooney, 1996, Zettlemoyer and Collins, 2005, Suhr et al., 2018]. The Python API of our environments is based on the semantic parsing work of Goldman et al. [2018]. Robust semantic parsers can assist in automating our annotation process.

