# OpenReview forum: "${lil}$Gym: Natural Language Visual Reasoning with Reinforcement Learning"
_NeurIPS.cc/2022/Workshop/LaReL — LaReL 2022_

### Official Review · Reviewer_1CsD · 2022-10-16

**Rating:** 7
**Confidence:** 4

**Review:**

This paper describes a new suite of reinforcement learning problems derived from the NLVR visual sentence classification dataset of Suhr et al. Here, rather than simply recognizing whether a natural language statement is true of a given image, the learner's job is to manipulate the image in order to make the statement true. Two forms of manipulation are considered: in one, the agent can simply add or remove blocks from a stack; in the other, the agent can freely place or delete objects on a 2-D grid. Evaluated in these environments, two standard vision-and-language RL baselines complete a substantial fraction of tasks, while remaining far from solving all of them.

I think this is an interesting environment, has the potential to be useful for other researchers' ongoing work, and would be a great fit for the workshop. Here are a couple of directions that would be interesting to explore if the authors want to continue working in this direction toward a complete conference submission:

- More fine-grained analysis of *what* is difficult about these environments. Are there structural differences between the tasks that the agent solves and fails to solve? Particular visual / relational / compositional concepts that are particularly challenging or easy?

- Some discussion of what's different about the RL version of this task versus the image classification version. Again, is difficulty (measured at the instance level) highly correlated between task formulations, or are there particular behaviors / concepts that are easy to learn in the supervised setting, but hard to learn via RL?

- Finally, since you have the program annotations, what can we say about the relationship between programs and learned behaviors from the RL agent? Can we, for example, probe the agent's visual representations to get out programs (or particular features of those programs)?

It would also be great to include more examples from the dataset. The example in Fig 1 is trivial, and Fig 2 in the appendix does a much better job of highlighting what makes this task interesting / hard.

---

### Official Review · Reviewer_n6si · 2022-10-17
**Introduces a useful benchmark; well-executed experiments**

**Rating:** 7
**Confidence:** 4

**Review:**

Summary: This paper introduces a new benchmark for natural language visual reasoning. The authors introduce different configurations, like tower and scatter, and show that some baseline models achieve good performance with room for improvement.

Strengths:
- The paper addresses an important problem by introducing a dataset to make language+RL+reasoning research more effective.
- The paper is well-written, and easy to follow.
- The experiments are well-executed, with insightful analysis.

---

### Decision · Program_Chairs · 2022-10-21

Accept